# Understanding primary care diagnosis and management of sleep disturbance for people with dementia or mild cognitive impairment: a realist review protocol

Leanne Greene ,[1] Aidin Aryankhesal,[2] Molly Megson,[3] Jessica Blake,[2] Geoff Wong,[4] Simon Briscoe,[1] Andrea Hilton,[5] Anne Killett ,[2] Joanne Reeve,[3] Louise Allan,[1] Clive Ballard,[1] Niall Broomfield,[6] Jayden van Horik,[1] Mizanur Khondoker,[6] Alpar Lazar,[2] Rachael Litherland,[7] Gill Livingston,[8] Ian Maidment ,[9] Antonieta Medina-Lara,[1] George Rook,[10] Sion Scott,[11] Lee Shepstone,[6] Chris Fox[1]

**Correspondence to**
Dr Leanne Greene;
l.m.g.greene@exeter.ac.uk

## ABSTRACT

**Introduction** The increasingly ageing population is associated with greater numbers of people living with dementia (PLwD) and mild cognitive impairment (MCI). There are an estimated 55 million PLwD and approximately 6% of people over 60 years of age are living with MCI, with the figure rising to 25% for those aged between 80 and 84 years. Sleep disturbances are common for this population, but there is currently no standardised approach within UK primary care to manage this. Coined as a 'wicked design problem', sleep disturbances in this population are complex, with interventions supporting best management in context.

**Methods and analysis** The aim of this realist review is to deepen our understanding of what is considered 'sleep disturbance' in PLwD or MCI within primary care. Specifically, we endeavour to better understand how sleep disturbance is assessed, diagnosed and managed. To co-produce this protocol and review, we have recruited a stakeholder group comprising individuals with lived experience of dementia or MCI, primary healthcare staff and sleep experts. This review will be conducted in line with Pawson's five stages including the development of our initial programme theory, literature searches and the refinement of theory. The Realist and Meta-narrative Evidence Syntheses: Evolving Standards (RAMESES) quality and reporting standards will also be followed. The realist review will be an iterative process and our initial realist programme theory will be tested and refined in response to our data searches and stakeholder discussions.

**Ethics and dissemination** Ethical approval is not required for this review. We will follow the RAMESES standards to ensure we produce a complete and transparent report. Our final programme theory will help us to devise a tailored sleep management tool for primary healthcare professionals, PLwD and their carers. Our dissemination strategy will include lay summaries via email and our research website, peer-reviewed publications and social media posts.

## STRENGTHS AND LIMITATIONS OF THIS STUDY

⇒ The realist review explores an under-researched area, namely what sleep disturbances mean for people living with dementia (PLwD) or mild cognitive impairment (MCI) and their carers, and how it is managed by primary healthcare professionals.

⇒ The realist approach will enable us to make sense of complex situations, helping to identify why and when there may be variations in the experience of sleep disturbance between PLwD or MCI and in primary care responses.

⇒ We will collaborate with stakeholders to ensure the programme theory makes sense and is relevant to people using it.

⇒ Sleep disturbance and its management in primary care for PLwD or MCI is complex, and the final programme theory may only present partial knowledge, which will improve with future research.

**PROSPERO registration number** CRD42022304679.

## INTRODUCTION

Worldwide, approximately 55 million individuals have a dementia diagnosis.[1] Global prevalence predictions suggest that the number of people living with dementia (PLwD) will rise to 78 million by 2030, and to 139 million by 2050.[2] Rising prevalence rates are associated with increasing human life expectancy[2] and the heightened risk of those in the oldest age brackets (ie, 80 years and above) developing dementia.[3] Dementia has major economic, healthcare and societal costs,[4 5] with predictions estimating that it will be the most expensive chronic condition in the future. In 2020,

figures indicated a global cost of approximately US$1.3 trillion. This is anticipated to double by 2050.[6]

Although not definitive, mild cognitive impairment (MCI) can be broadly defined as minimal cognitive impairment, measured objectively through neuropsychological tests or subjectively through self or respondent reports, that does not significantly impact activities of daily living (ADL) or instrumental functions.[6] The diagnostic criterion of MCI is complex and is dependent on which assessments are used, the definition of ADLs, population norms and the person's premorbid cognitive levels.[7] Due to variations in definitions, diagnostic approaches and a general paucity of national and international guidance, it is hard to accurately predict the prevalence rates of MCI.[7] Review data indicate that approximately 6% of those over 60 years of age are living with MCI, with the figure rising to 25% for those aged between 80 and 84 years.[8]

Sleep disturbances are common for PLwD (20%–90%) and MCI (18.3%–45.5%).[9–13] Problems with sleep can involve disturbance to the quantity, quality and timing of sleep, and can stem from physical or psychological conditions.[14] They can be exacerbated by factors such as age-related changes in the circadian rhythm, medication side effects and comorbidities.[15] Sleep disturbance in dementia and MCI populations has been associated with poorer daily functioning,[16 17] and can significantly impact caregivers.[18] From a primary care perspective, sleep assessment and management in dementia and MCI populations is complex, requiring a multidisciplinary understanding and tailored approach that considers individualised priorities and actions. Finding a clinically effective and safe way to manage sleep disturbances for PLwD and MCI remains a challenge.[19] The prescription of licensed sleep medications is complex and may be associated with adverse events, although these are still prescribed.[20] Emerging evidence indicates that non-pharmacological interventions may be beneficial.[21 22] From our previous work, PLwD and those with MCI fear 'abandonment' if primary care clinicians reduce or stop sleep medication. Therefore, sleep management presents as a 'wicked design problem' because the issue is complex, the knowledge required to understand and address it includes multiple disciplines and subject expertise and interventions do not 'fix' but rather support best management in context.[23 24] The issue is also 'wicked' in that its potential solutions can create other problems.[25]

### Rationale explaining why this research is important

Both the current literature[17 26 27] and consultations with our stakeholders highlight the importance of investigating techniques that alleviate sleep disturbance among PLwD or MCI. Following the Medical Research Council's framework for developing and evaluating complex interventions,[28] this review is the first work package of a larger research programme that aims to gather data to better understand sleep disturbance in dementia and MCI populations. The review will report on assessments, interventions and management used for sleep disturbance for PLwD and MCI. Findings will describe the 'active ingredients' of the aforementioned, how they were implemented, what measures were used, to what extent they worked, for whom and why. We know that current theoretical models and guidance are not sufficient to support complex, tailored primary care.[29] The findings from this review are needed to refine the intervention components necessary to support individually tailored optimisation of care in the context of sleep disturbance in dementia and MCI populations. In the context of person-centred primary care practice, management of complex problems, including sleep disturbance, is not just about (correct) 'diagnosis', but creating and using a tailored explanation and understanding of the problem as a basis for management. This review seeks to understand this complex process.

### Review objectives and design

This review protocol has been registered with PROSPERO and will follow current quality and publication standards.[30]

To ensure that the focus of our research was relevant to stakeholders, we collaborated in patient and public involvement (PPI) consultations with 12 carers from Together in Dementia Everyday and 10 PLwD or people with MCI from the Dementia Engagement and Empowerment Project. The PPI representatives told us that sleep disturbances are a major priority and that they wanted further help with this. We have used this feedback to develop the aims, objectives and research questions for this review and continue to work with our PPI collaborators including the Alzheimer's Society and Chinese Well-being. One aim of the wider TaIlored ManagEment of Sleep (TIMES) programme is to work closely with the Chinese and South Asian communities to strengthen the reach and implementation of the intervention.

A realist review methodology was chosen because the aim of the overall TIMES programme is to develop and evaluate a complex sleep intervention for PLwD, MCI and health professionals in primary care. The intervention is likely to have a range of outcomes for different groups of people as well as being context-sensitive. This methodology will allow us to adopt a theory-driven approach to synthesising data and offering findings that clearly describe how and why context can influence outcomes.

### Review aims

The aim of this review is to gain a better understanding of what is considered 'sleep disturbance' in dementia/MCI from a whole-person perspective. We aim to better understand how this is assessed, diagnosed and managed within primary care. Our findings will help to devise a tailored sleep management tool to be used by primary care health professionals, PLwD/MCI and their carers.

### Review objectives

1. To conduct a realist review to better understand what sleep disturbance means to PLwD and MCI, their

carers and primary healthcare professionals assessing/managing this.

2. To explore how sleep disturbances are assessed, diagnosed and managed for PLwD/MCI who live in the community. We are particularly interested in the 'active ingredients' of assessment, diagnosis and management, how they are implemented, what measures are used, to what extent they worked, for whom and why.

3. To synthesise the findings into a realist programme theory which will refine the description of the core components of a tailored sleep management tool. Through an iterative consultation process, we will present findings from the realist review to describe the ideal components of tailored care to a co-production group, including those with lived experience of dementia and MCI.

### Review questions

1. How do clinicians in primary care deliver tailored diagnosis and management of sleep disturbances for PLwD or MCI who live in the community?

2. When are certain management strategies used, why, by and for whom and to what extent?

3. How do PLwD/MCI and unpaid family carers report or influence the assessment, diagnosis and management of sleep disturbances?

The review has and will continue to be conducted in line with Pawson's five stages[31 32] including the development of our initial programme theory, literature searches and the refinement of theory. The Realist and Meta-narrative Evidence Syntheses: Evolving Standards (RAMESES) quality[33] and reporting[30] standards will also be followed.

### Stakeholder involvement

We have created a diverse national and international stakeholder group with both content and lived experience expertise. This group includes, for example, expert sleep clinicians, primary care clinicians, Electronic Checking Leading to Improved Prescribing Safety and Efficiency representatives, PPI through a Lived Experience Advisory Forum for Sleep and relevant charities. We have worked with the TIMES stakeholder group to focus our review questions and we will collaborate with them to refine our initial programme theories and to provide feedback on the iterative versions of the programme theory, ensuring they are relevant to their experiences.[34] We will meet on four occasions (either virtually or face-to-face) throughout the project and correspond as necessary via email. A PPI newsletter will be sent with updates and encouragement to engage in the project. Stakeholders will have the opportunity to provide feedback on project materials, findings and recommendations, and to support with dissemination.

### Patient and public involvement

PPI is a key factor in the delivery of the TIMES research programme. We have a PPI co-applicant who is living with dementia and is a vital member of the research team. Our PPI members ensure that PLwD and carers' voices are kept at the forefront of the research, shaping the design of the research and the interpretation and dissemination of findings. TIMES strives to conduct inclusive research to combat health inequalities, so we are actively seeking representatives from underserved groups.[34] We are collaborating with ethnic minority carer and patient groups, such as Chinese Well-being, people who live alone with dementia and people who are in the mid-stage or later stage of dementia and may lack capacity. PPI co-involvement will help us focus the review, test the programme theory and help us to comprehend important contexts.

### Step 1: develop initial programme theory

The first stage of our realist review was the development of an 'initial programme theory'. To achieve this, we conducted an exploratory review of the literature around the management of sleep and tailored care in the context of primary care and dementia/MCI. We sought stakeholder input. From these activities, we drafted an initial programme theory describing the clinical work which will be involved in the development of a tailored approach to managing sleep disturbances in PLwD or MCI (figure 1). This was based on an existing understanding of (a) the problem to be addressed (improving the management of persistent sleep problems in dementia/MCI through tailored assessment); (b) barriers to tailored care and (c) evidence-informed approaches that may overcome those barriers. Our review will highlight gaps in our knowledge on how to assess, diagnose and manage sleep problems in the community, and help us to address these deficits in knowledge. As the review progresses, the emerging programme theory will be used to stimulate depth of empirical exploration in future work packages. The empirical work will then further inform the realist synthesis data interpretation.

### Step 2: searching for evidence

Primary and secondary evidence will be routinely collected during the development of the realist review and will be based on an iterative interpretation of confirming, refuting or refining different aspects of the programme theory.[35] Searches will be developed and revised as needed with an information specialist and stakeholder input from members of the TIMES stakeholder group. The hierarchy is not the traditional hierarchy as is typical when exploring evidence for the effectiveness of interventions, but about data that give insights into complexity. Therefore, it is anticipated that there will be a wide range of sources including, but not limited to, quantitative and qualitative data, peer-reviewed articles, commentaries and grey literature.[33] We anticipate using the following databases: MEDLINE, PsycINFO, CINAHL, ASSIA, Health Management Information Consortium and OpenGrey. Free text and subject heading search terms will be chosen by the information specialist and discussed and refined (if needed) with the research team and stakeholders to ensure a balance between sensitivity and specificity.[36]

**Figure 1. Initial programme theory for the assessment, diagnosis, and management of sleep disturbances for people with dementia or mild cognitive impairment living in the community**

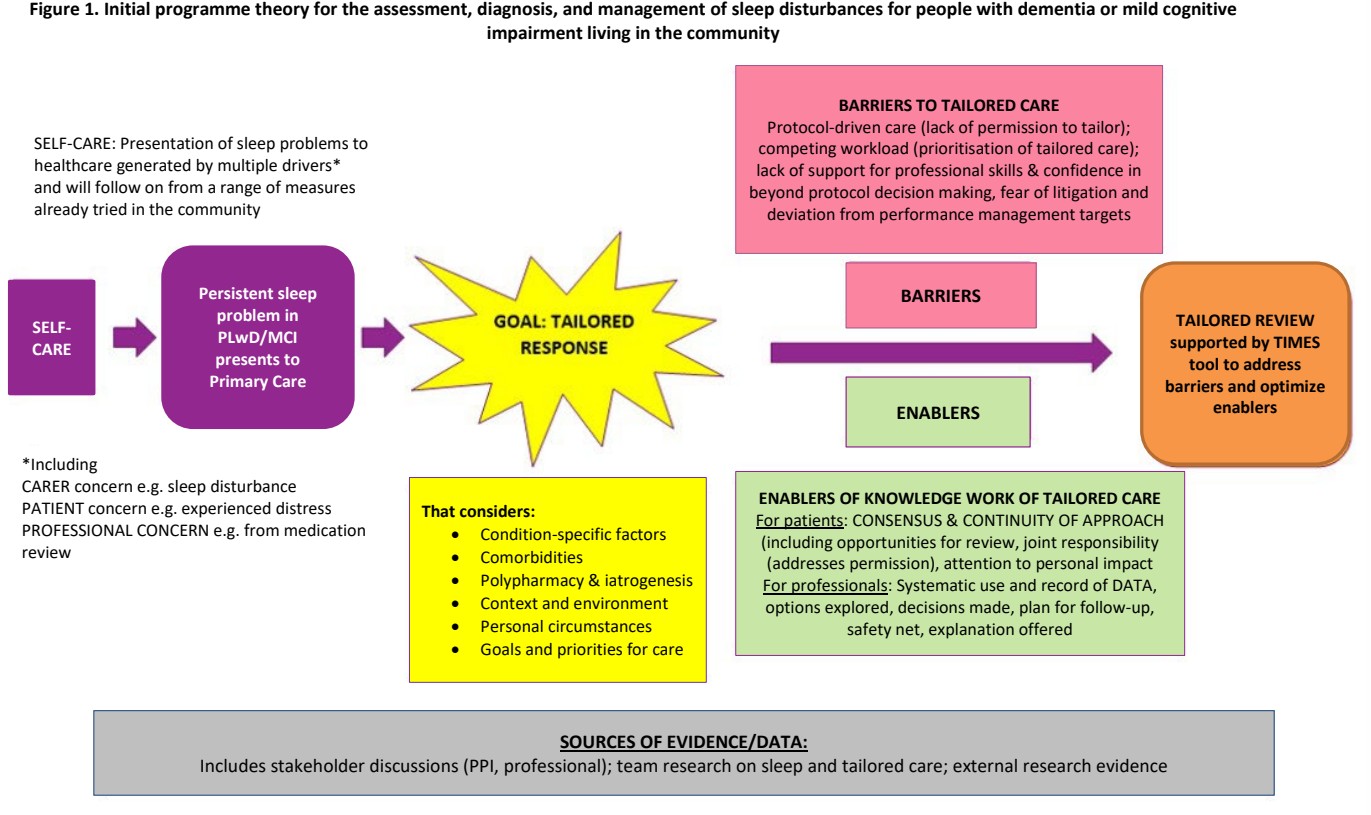

**Figure 1** Initial programme theory for the assessment, diagnosis and management of sleep disturbances for people living with dementia (PLwD) or mild cognitive impairment (MCI) living in the community. PPI, patient and public involvement.

Additional documents will be sought through citation tracking and approaching topic experts.[37] Grey literature will also be captured through web searching.

The inclusion and exclusion criteria will be used to screen the results.

### Inclusion criteria

Population: PLwD, individuals with MCI, informal, unpaid, family/friend carers and/or primary/community healthcare professionals.

Phenomenon of interest: sleep disturbance.

Context: primary care.

Study design/types of documents: peer-reviewed pharmacological and non-pharmacological research studies and clinical guidelines (any design, eg, quantitative, qualitative and mixed methods evaluations), policy documents, local and non-clinical guidelines, review studies (eg, Cochrane reviews, systematic reviews, umbrella reviews), websites of professional bodies and conference abstracts.

### Exclusion criteria

Population: participants in hospital, secondary, tertiary, hospice care and/or participants with life expectancy of 3 months or less.

Context: studies, reviews or data based in hospitals, secondary, tertiary, hospice care or non-Organisation for Economic Co-operation and Development nations.

Study design/types of documents: studies, reviews, data including end-of-life care (life expectancy of 3 months or less). Information not published in English.

Searches will be screened against the inclusion criteria in three steps: first by title, abstract and keywords, and second by full text. Screening will be undertaken by three of the TIMES researchers, and each will check a 10% random sample at each step to look for systematic errors. The three screeners will meet when needed to discuss discrepancies and agreements and resolve these through discussion. If any disagreements remain, they will be brought to the wider review team for discussion. If any further disagreements remain voting will be used and a majority decision will be taken. Supplementary searches may be necessary if the preliminary search does not generate adequate data to test the programme theory. These searches will follow the same screening process as described above. The outcome of the search and screening process will be reported in full, including Preferred Reporting Items for Systematic Reviews and Meta-Analyses style flow diagrams.[38]

### Step 3: selection and appraisal

During screening, documents will be finally selected for inclusion based on their relevance to the review topic (ie, if the document has information which could test, ie, confirm, refute or refine the programme theory)

and rigour (ie, that the data have sufficient integrity to warrant changes to the programme theory).[32] It should be noted that evaluating rigour during a realist review differs from traditional reviews.[30 37 39] We will mainly judge each piece of data according to its relevance for testing the programme theory.[32] We will mostly make rigour judgements at the level of plausibility and coherence of the programme theory using the criteria of consilience, simplicity and analogy.[30] The processes in this step will be undertaken by one member of the review team with regular input and support from the wider project team.

## Step 4: extracting and organising data

Characteristics of included documents will be extracted into an Excel spreadsheet by the review team. Documents included in the review will be uploaded into NVivo QRS international qualitative data analysis tool to facilitate data analysis.[40] Deductive coding (directed by the initial programme theories) and inductive coding (created while reading papers) will be performed to organise and classify the data.[32] Data within the codes will be used to develop and then confirm, refute or refine the context-mechanism-outcome (CMO) configurations that are contained within the initial programme theory. Data analysis will involve the use of a realist logic of analysis with the goal of using data from the literature to further develop the initial programme theory. Analysis requires interpretation and judgement of data. We will use a series of questions to guide the interpretations and judgements about the sources, moving iteratively between the analysis of particular examples and refinement of the programme theory. As the programme theory evolves, we will seek input from members of the TIMES stakeholder and PPI group. Data used to refine the programme theory will be reported transparently.[32] We have used this process of analysis successfully in other realist reviews.[36 41] A second reviewer will randomly review a 10% sample of documents that have been through the data extraction and organisation process to ensure consistency.

## Step 5: analysis and synthesis

During a realist review, analysis and synthesis simultaneously occur throughout several stages. The process will start at the same time as document selection and appraisal (step 3) and will continue throughout data extraction and organisation (step 4) in accordance with the data's potential to alter the existing programme theory. This process is iterative, and the programme theory will continue to develop as more data are analysed. Stakeholder input will be sought during this stage and if there are any gaps in the data further searches will be conducted. In line with Pawson's guidance,[32] data linking to different areas of the programme theory will be gathered together and analysed alongside each other and retroductive reasoning will be used to enable us to develop CMO configurations. The retroductive reasoning requires iterative movements between concrete observations (ie, inductive reasoning) and theory building (ie, deductive reasoning).[42] Doing so will enable us to interpret which underlying causal mechanisms must be present to obtain the observed patterns of outcomes.

## Limitations and risks

Sleep disturbance is an umbrella term for many different conditions and symptoms. There is also an element of subjectivity in how people experience or interpret sleep disturbances. As there are limitations on how much a single realist review study can cover, the review will focus on the aspects of sleep disturbance that we judge to be the most important to initially address,[42] meaning that some aspects may have to be set aside. Therefore, one limitation of the current research is that the final programme theory will only present a partial knowledge of the complex sleep disturbance issue. However, future research will refine and refute the published programme theory to create a more well-rounded theory.

## Outputs and dissemination

The RAMESES reporting standards will be followed to ensure our review is transparent.[30] There will be a stand-alone peer-reviewed publication for academic audiences as well as scope for conference presentations. Findings will be presented to the stakeholder group via online meetings and email correspondence. Stakeholders will also be able to advise us on our dissemination strategy to ensure we have a wide distribution of findings to people interested in the development and evaluation of interventions for PLwD/MCI who experience sleep disturbances. Accessible summaries of findings will also be presented on the TIMES website and Twitter account.

**Author affiliations**
[1]Clinical Trails Unit, University of Exeter Medical School, University of Exeter, Exeter, UK
[2]Faculty of Medicine and Health Sciences, School of Health Sciences, University of East Anglia, Norwich, UK
[3]Academy of Primary Care, Hull York Medical School, University of Hull, Hull, UK
[4]Nuffield Department of Primary Care Health Sciences, Oxford University, Oxford, UK
[5]Faculty of Health Sciences, School of Paramedical PeriOperative and Advanced Practice, University of Hull, Hull, UK
[6]Faculty of Medicine and Health Sciences, Norwich Medical School, University of East Anglia, Norwich, UK
[7]Innovations in Dementia, Exeter, UK
[8]Faculty of Brain Sciences, Division of Psychiatry, University College London, London, UK
[9]College of Health and Life Sciences, Aston Pharmacy School, Aston University, Birmingham, UK
[10]Living with Dementia, N/A, UK
[11]College of Life Sciences, School of Allied Health Professions, University of Leicester, Leicester, UK

**Acknowledgements** The authors would like to thank all stakeholders and PPI members for their contribution to the research.

**Contributors** The idea for the review was conceived as part of the wider NIHR202345 funding with inputs from CF, GW, LG, AH, JR, AK, AA, MM, IM, LA, CB, SS, NB, GL, GR, AM-L, JB, AL, LS, RL and MK. LG prepared the protocol as a manuscript for publication with input from GW, CF, AH, JR, AK, AA, MM, IM, LA, SS, NB, GL, GR, SB, AM-L, JB, AL, CB, LS, RL, MK, JvH. All authors contributed to

subsequent revisions and approved the protocol prior to its submission. CF is the guarantor.

**Funding**  This project is funded by the National Institute for Health and Care Research (NIHR) Programme Grants for Applied Research (NIHR202345).

**Competing interests**  None declared.

**Patient and public involvement**  Patients and/or the public were involved in the design, or conduct, or reporting, or dissemination plans of this research. Refer to the 'Methods' section for further details.

**Patient consent for publication**  Not applicable.

**Ethics approval**  Not applicable.

**Provenance and peer review**  Not commissioned; externally peer reviewed.

**ORCID iDs**
Leanne Greene http://orcid.org/0000-0002-5383-4362
Anne Killett http://orcid.org/0000-0003-4080-8365
Ian Maidment http://orcid.org/0000-0003-4152-9704

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
