## [Reviewer comments · BMJ Open]

ARTICLE DETAILS

TITLE (PROVISIONAL)	Understanding primary care diagnosis and management of sleep disturbance for people with dementia or mild cognitive impairment: A realist review protocol
AUTHORS	Greene, Leanne; Aryankhesal, Aidin; Megson, Molly; Blake, Jessica; Wong, Geoff; Briscoe, Simon; Hilton, Andrea; Killett, Anne; Reeve, Joanne; Allan, Louise; Ballard, Clive; Broomfield, Niall; van Horik, Jayden; Khondoker, Mizanur; Lazar, Alpar; Litherland, Rachael; Livingston, Gill; Maidment, Ian; Medina-Lara, Antonietta; Rook, George; Scott, Sion; Shepstone, Lee; Fox, Chris

VERSION 1 – REVIEW

REVIEWER	Camargos, Einstein Francisco
REVIEW RETURNED	20-Sep-2022

GENERAL COMMENTS	Dear Editor, thanks for the invitation to review the protocol titled "Understanding primary care diagnosis and management of sleep disturbance for people with dementia or mild cognitive impairment: A realist review protocol." The authors propose conducting a review article on sleep disturbance in cognitively impaired patients within primary care (assessment, diagnosis, and management). There are scarce studies approaching sleep disorders in cognitively impaired people, especially in primary care settings. The method to conduct this study is a "Realist and Meta-narrative review," developed to help make sense of heterogeneous evidence about complex interventions applied in diverse contexts in a way that informs policy. It is appropriate to the topic studied (sleep disorders in the primary setting). The protocol is well written, and the authors appropriately followed the list of items to be included when reporting a meta-narrative review (RAMESES publication standards: meta-narrative reviews). I have a few comments below: Page 4. Line 23-26. "MCI can be broadly defined as one or more cognitive impairments examined via objective neuropsychological assessments that diminish participation in activities of daily living (ADLs)". Minimal impairment in complex instrumental functions is more correctly used to define MCI. Page 4. Line 51-53. "PLwD and those with MCI fear 'abandonment' if primary care clinicians reduce or stop sleep medication". In my opinion, it is crucial in primary care. Page 6. Review questions 3. It is another crucial point. A study conducted by McCurry and cols showed that, although caregivers report all patients to have multiple sleep disturbances, 41% of patients had actigraphic sleep efficiencies in the normal range and
--

	43% averaged eight or more hours of sleep nightly (Am J Geriatr Psychiatry 2006; 14:112–120). Thus, setting accurate instruments to diagnose SD is essential. In addition, in my experience, when caregivers are confronted with actimetric data demonstrating adequate sleep, they change their minds about their sleep patterns. A study found that caregivers of patients with dementia took psychotropic drugs (benzodiazepines and antidepressants) more frequently than the ones of patients without dementia (Arq Neuropsiquiatr 2012;70(3):169-174). I would suggest caregivers as a vital part of this protocol study.
--	---

VERSION 1 – AUTHOR RESPONSE

Comment	Response	Location of change
The protocol is well written, and the authors appropriately followed the list of items to be included when reporting a meta-narrative review (RAMESES publication standards: meta-narrative reviews).	Thank you for your positive response to our paper and for the thoughtful feedback.	N/A
Page 4. Line 23-26. "MCI can be broadly defined as one or more cognitive impairments examined via objective neuropsychological assessments that diminish participation in activities of daily living (ADLs)". Minimal impairment in complex instrumental functions is more correctly used to define MCI.	We have amended the definition, so it reads 'Although not definitive, mild cognitive impairment (MCI) can be broadly defined as minimal cognitive impairment, measured objectively through neuropsychological tests or subjectively through self/respondent report, that does not significantly impact activities of daily living or instrumental functions.'	Page 4, lines 15-18
Page 4. Line 51-53. "PLwD and those with MCI fear 'abandonment' if primary care clinicians reduce or stop sleep medication". In my opinion, it is crucial in primary care.	Thank you for this insight, we really appreciate it. We will keep this in mind when we analyse our data as it is a valid point.	N/A
Page 6. Review questions 3. It is another crucial point. A study conducted by McCurry and cols showed that, although caregivers report all patients to have multiple sleep disturbances, 41% of patients had actigraphic sleep efficiencies in the normal range and 43% averaged eight or more hours of sleep nightly (Am J Geriatr Psychiatry 2006; 14:112–120 PubMed). Thus, setting accurate instruments to diagnose SD is essential. In addition, in my experience,	Thank you for both comments - the value of accurate diagnoses for sleep problems and the importance of carers in the management of sleep problems for PLwD/MCI. We will ensure we consider the points you have raised as we analyse our data.	N/A

when caregivers are confronted with actimetric data demonstrating adequate sleep, they change their minds about their sleep patterns. A study found that caregivers of patients with dementia took psychotropic drugs (benzodiazepines and antidepressants) more frequently than the ones of patients without dementia (Arq Neuropsiquiatr 2012;70(3):169-174). I would suggest caregivers as a vital part of this protocol study.		
---	--	--